# Characterizing Vision Backbones for Dense Prediction with Dense Attentive Probing

**Timo Lüddecke and Alexander Ecker**

**Reviewed on OpenReview:** [https://openreview.net/forum?id=neMAx4uBlh](https://openreview.net/forum?id=neMAx4uBlh)

## Abstract

The paradigm of pretraining a backbone on a large set of (often unlabeled) images has gained popularity. The quality of the resulting features is commonly measured by freezing the backbone and training different task heads on top of it. However, current evaluations cover only classifications of whole images or require complex dense task heads which introduce a large number of parameters and add their own inductive biases. In this work, we propose dense attentive probing, a parameter-efficient readout method for dense prediction on arbitrary backbones – independent of the size and resolution of their feature volume. To this end, we extend cross-attention with distance-based masks of learnable sizes. We employ this method to evaluate 18 common backbones on dense predictions tasks in three dimensions: instance awareness, local semantics and spatial understanding. We find that DINOv2 outperforms all other backbones tested – including those supervised with masks and language – across all three task categories. Furthermore, our analysis suggests that self-supervised pretraining tends to yield features that separate object instances better than vision-language models. Code is available at [https://eckerlab.org/code/deap](https://eckerlab.org/code/deap).

## 1 Introduction

Driven by the success of self-supervised learning (Chen et al., 2020; He et al., 2022; Oquab et al., 2023) and vision-language training (Radford et al., 2021) training from scratch has largely been replaced by fine-tuning large pretrained backbones for many computer vision tasks. Ideally, the pretraining yields powerful features such that fine-tuning succeeds even with small datasets and by modifying only a small number of parameters. Many factors influence the feature quality of pretrained backbones, including the pretraining paradigm, model architecture and the training data. Therefore, for both computer vision scientists and practitioners, it is crucial to characterize strengths and weaknesses of large pretrained backbones through systematic benchmarks. For assessing the whole-image classification performance of backbones (i. e. predicting one label per image) such benchmarks exist, for example, ImageNet (Russakovsky et al., 2014), VTAB (Zhai et al., 2019) and FGVC (Jia et al., 2022). They often rely on the established approaches of linear and attentive probing. Linear probing applies global average pooling and then linearly maps the resulting feature vector to class predictions while attentive probing uses cross-attention with a query token that predicts the class. Both techniques are not applicable for producing dense output due to limited resolution. For example, in linear probing the prediction has the same spatial resolution as the feature volume, which is usually too low to capture object structures. Furthermore, the resolution of the feature volume varies across different backbones, preventing a fair comparison. Therefore, often common task heads such as UPerNet (Chen et al., 2024a) for semantic segmentation or Faster-RCNN for object detection (Ren et al., 2015) are used at the price of introducing a large number of learnable parameters and additional inductive biases. As a consequence, the resulting backbone feature quality measurements are mediated by compatibility and performance of these heads.

In this work, we address the problem of assessing and comparing the representational quality of dense feature volumes from various backbones. To this end, we measure dense prediction performance of backbones as directly as possible by proposing a novel dense equivalent to attentive probing. Our model consists of a

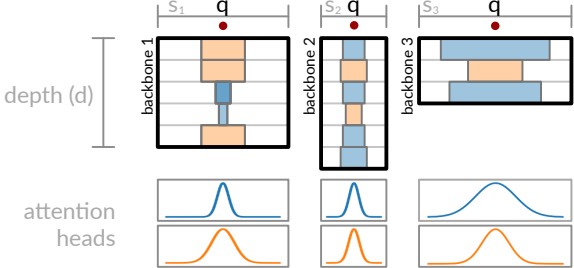

Figure 1: Feature volumes of different backbones (black rectangles) differ in feature depth ($d$) and spatial size ($s$). We assume that the region size of informative features for each location $q$ (blue and orange rectangles) varies across backbones. To compensate for this, our model uses attention masks with learnable region size (blue and orange lines) per attention head and decouples feature and output resolution.

Table 1: Adapter methods for dense prediction with the number of learnable parameters.

| Model | learnable parameters |
|---|---|
| ConvAdapter (Chen et al., 2024a) | >24M |
| ViT-Adapter (Chen et al., 2023) | 2.5M - 23.7M |
| Conv LoRA (Zhong et al., 2024) | $\sim$ 4M |
| Dense Attentive Probing (ours) | 0.05M - 0.5M |

single masked cross-attention layer, introduces only a small number of parameters (less than 100K, often even less than 50K) and adds a relatively small computational overhead over the backbone. By using cross-attention, we decouple the size and resolution of the input image and encoder output from that of the dense output, i. e. generate outputs at any resolution. As the backbones that we evaluate were trained with different techniques (e.g. language-vision or masked image modeling), they might encode local information differently (Fig. 1). For example, information for predicting at output location $q$ might not exclusively be encoded at the corresponding location $p$ in the feature volume but be distributed over a large region around $p$ in the feature volume. This region can be defined as the set of spatial $(i, j)$ and feature ($d$) indices $(i, j, d) \in I(q)$ of the feature volume which contains relevant information to predict a target function (such as a segmentation mask) at location $q$. Ideally, a probing model would capture this region $I(q)$. Our model makes the simplifying assumption to define the region $I(q)$ smoothly by a Gaussian function which is used as attention mask in the cross-attention mechanism. By learning individual standard deviations of this function per attention head, the model can process different degrees of feature locality (or region sizes) in each dimension.

Balestriero et al. (2023) note the lack of a standardized evaluation protocol for self-supervised learning methods on dense prediction tasks. We believe our dense attentive probing method can address this gap and experimentally assess the quality of various supervised, self-supervised, vision-language training methods. Specifically, we use our dense probing method to characterize features for dense tasks along these three dimensions: (1) instance disentanglement, i. e. how well are individual instances recognizable from the features; (2) local semantics, evaluating how meaningful the features are for a local classification; and (3) spatial understanding, which assesses how well is the 3D structure of the scene captured. Our method is better suitable for evaluation than prior work because it decouples feature and output resolution, adapts feature locality and uses only a small parameter budget.

## 2 Related work

**Representation learning.** Self-supervised representation learning has been a popular research topic with multiple approaches that can roughly be categorized into joint-embedding (Chen et al., 2020; 2021; Caron et al., 2021) and reconstruction-based (He et al., 2022). DINOv2 is based on the iBot (Zhou et al., 2022) method which uses a joint-embedding architecture in combination with self-distillation and reconstruction. VicRegL (Bardes et al., 2022) and many other recent methods explicitly addresses local features by modeling losses at the token level. There has been a discussion about which techniques lead to better and more efficient features for perceptual tasks (Balestriero & LeCun, 2024). The approaches discussed above mainly address classification. Another stream of research, called object-centric learning, focuses on learning disentangled object representations. While early methods only worked on synthetic data (Burgess et al., 2019; Locatello

et al., 2020) more recent approaches succeed on natural images (Zadaianchuk et al., 2024; Aydemir et al., 2023). Recently, a new method for evaluating such object-centric representations was proposed Didolkar et al. (2025). The main difference of our present work to object-centric methods is that we assume that object instances are encoded implicitly in feature volumes whereas object-centric methods explicitly represent objects in their architecture (e. g. in slots). The seminal CLIP model (Radford et al., 2021) introduced another stream of research called vision-language models (VLMs), where the model is trained on aligning text-image pairs. Later, this training paradigm was simplified to use a sigmoid-based loss function (Zhai et al., 2023) instead of a softmax-based loss, making the method less dependent on the batch size. Recently, the role of data is investigated more closely in the context of vision-language models (Gadre et al., 2024; Xu et al., 2024; Fang et al., 2024b).

**Feature evaluation.** Evaluations on local features predate the deep learning era in computer vision, often addressing hard-coded descriptors which were computed around a sparse set of points of interest (Mikolajczyk & Schmid, 2005). Recently, there have been numerous attempts at characterizing and comparing features from common deep learning backbones. This involves evaluating the 3D understanding of backbones (El Banani et al., 2024; Man et al., 2024) as well as more specific aspects like physics (Zhan et al., 2024), language-binding (Lewis et al., 2024), compositionality (Thrush et al., 2022) or shape perception (Bonnen et al., 2024). Danier et al. (2025) compare numerous methods for depth estimation. Chen et al. (2024b) design a zero-shot benchmark for image encoders for the contrastive vision-language pretraining setting and propose the ViTamin architecture based on their findings. Goldblum et al. (2024) evaluate classification, instance segmentation, object detection and retrieval by training large task heads (Cascade Mask R-CNN) for dense prediction. Further efforts to characterize vision backbones include the timm leaderboard (Wightman, 2019) for image classification, CLIP benchmark (LAION-AI, 2022) for vision-language models, V-PETL for parameter-efficient fine-tuning (Xin et al., 2024) and CV-Bench for multimodal large language models (MLLMs; Tong et al., 2024). Our work differs in considering multiple pre-training paradigms (self-supervised, vision-language etc.) and multiple feature qualities (instance awareness, local semantics and spatial understanding) while fine-tuning with a small number of learnable parameters. Importantly, we avoid the use of standard task heads like UperNet (Chen et al., 2024a) for segmentation as they introduce own inductive biases and a large number of parameters which interferes with measuring the backbone's performance.

**Adapters and parameter-efficient fine-tuning methods.** Fine-tuning of large backbones using a small parameter budget is a well-studied problem under the umbrella term parameter efficient fine-tuning (PEFT). Our work shares the goal of keeping the number of parameters minimal. A common approach to address this problem is to use adapters (Houlsby et al., 2019; Hu et al., 2022; Lian et al., 2022), which fine-tune by adding learnable small, low-parameter sub-networks within larger backbones or fine-tune only on a subset of weights. While these techniques originate in natural language processing, they have recently been applied to computer vision. However, many adapters only address whole-image classification tasks Chen et al. (2022); Steitz & Roth (2024). Existing dense prediction adapters are not suitable for our goal of comparing backbones due to a large number of parameters: ViT-Adapter (Chen et al., 2023) builds on established task heads for segmentation (UperNet) and detection (Mask R-CNN and HTC++). The number of parameters introduced by this adapter depends on the backbone, ranging from 2.5M to 23.7M parameters. ConvAdapter (Chen et al., 2024a) propose an adapter specifically for convolutional networks. In case of dense prediction, their method uses standard task heads which require a large number of parameters. Zhong et al. (2024) apply low-rank adaptation to image segmentation tasks by introducing around 4M parameters. These methods require substantially more parameters than our method (Table 1). For an in-depth review of adapters we refer to the survey of Yu et al. (2024). In addition to adapters, there are other parameter-efficient fine-tuning techniques: Attentive probing (Yu et al., 2022; Bardes et al., 2023) is an inspiration for our work but is limited to whole-image classification in its original formulation. Oquab et al. (2023) use bilinear interpolation in the feature volume to make dense predictions. However, this approach is limited in its output resolution and unsuitable for a comparison as the feature volume sizes vary across backbones. Their advanced multi-scale readout (called "ms") combines activations from multiple layers, adopting this would complicate a comparison across backbones. FeatUp (Fu et al., 2024) is generic method to upscale feature volumes and thus also applicable here. We will use this method as a baseline.

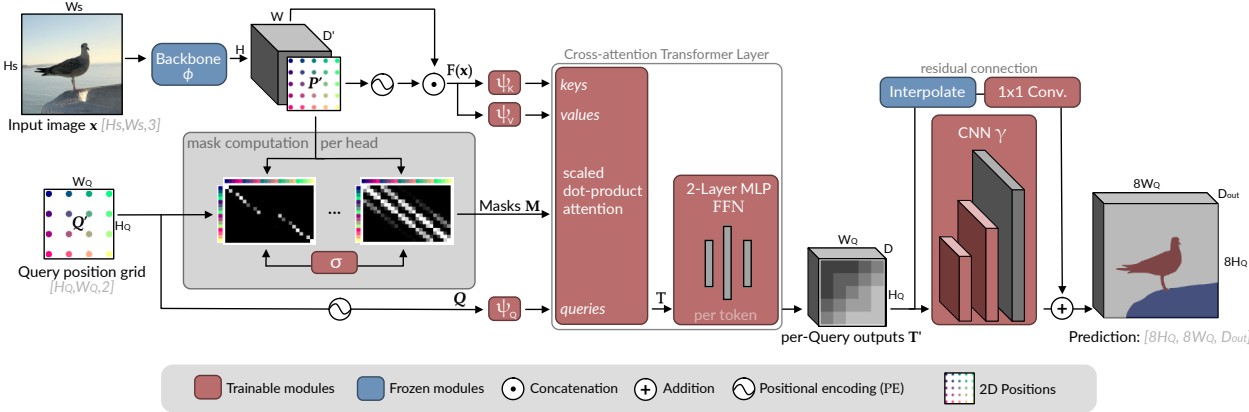

Figure 2: Dense attentive probing design: An input image $\mathbf{x}$ is processed by an arbitrary backbone $\phi$ and positionally encoded (see $\mathbf{P}'$), yielding the feature volume $\mathbf{F}(\mathbf{x})$. Queries ($\mathbf{Q}$) in form of a position-encoded coordinate grid cross-attend to these features $\mathbf{F}(\mathbf{x})$. Analogous to the Transformer layer, queries, keys and values are projected ($\psi$) before attending. The attention is constrained to a local region by a mask per head. These masks are computed from pairwise spatial distances between locations in the feature volume and in the query grid. The standard deviation $\sigma$ is a learnable scalar parameter that controls the size of the mask in each attention head. The resulting tokens are upscaled by a small CNN $\gamma$ to yield a task-specific output. Since the query position grid determines the output size, predicted output size and size of the feature volume are decoupled.

## 3   Dense Attentive Probing

In this section, we introduce the **De**nse **A**ttentive **P**robing (DeAP) method (Fig. 2). It is designed to be a parameter and compute efficient readout that uses cross-attention to make dense predictions based on features from a frozen backbone. Instead of allowing all queries to attend to the complete feature volume (full cross-attention), we constrain the attentive region locally such that each query can only interact with a local feature vector neighborhood. The size of this smooth local region is learnable to adapt to the characteristics of different backbones.

An arbitrary, frozen image backbone $\phi$ receives an image $\mathbf{x}$ of size $(Hs, Ws, 3)$ and generates features of size $(H, W, D')$ with $s$ indicating the backbone's stride (the factor by which the backbone reduces the spatial resolution of its input). These features are concatenated (denoted by [ ]) with a non-learnable, sinusoidal positional encoding (PE) of the grid coordinates $\mathbf{P}'$, yielding $\mathbf{F}(\mathbf{x})$ of size $(HW, D)$:

$$\mathbf{F}(\mathbf{x}) = [\text{PE}(\mathbf{p}), \phi(\mathbf{x})] \tag{1}$$

To generate a dense output, we use a cross-attention-based approach, where each query is responsible for generating output in a local region. The spatial queries $\mathbf{Q}$ of size $(H_Q W_Q, 16)$ are fixed (non-learnable), 8-dimensional positional sinusoidal encodings (Vaswani et al., 2017) of the (output) position grid. This position grid consists of integer positions, i.e. $Q_{ij} = [\text{PE}(i), \text{PE}(j)]$. The queries $\mathbf{Q}$ interact with the feature volume $\mathbf{F}(\mathbf{x})$ using a cross-attention mechanism. To enable the model to account for feature locality, we modify the cross-attention to consider spatial proximity by adding $\mathbf{M}(\sigma)$ to the attention scores. The computation for each head $h$ (out of $H$ heads) is described by

$$\mathbf{T}^{(h)} = \text{softmax}\left(\frac{\psi_Q^{(h)}(\mathbf{Q})\psi_K^{(h)}(\mathbf{F}(\mathbf{x}))^T}{\sqrt{\dim_k}} + \mathbf{M}(\sigma)\right)\psi_V^{(h)}(\mathbf{F}(\mathbf{x})), \tag{2}$$

with the functions $\psi_{\{V,K,Q\}}^{(h)}$ flattening along the spatial dimensions followed by multiplications with matrices of sizes $(D, \frac{2D}{H})$, $(D, \frac{D}{H})$ and $(8, \frac{D}{H})$ for values (V), keys (K) and queries (Q) respectively.

| | Supervised | | | Self-supervised | | | | | Vision-language | | |
|---|---|---|---|---|---|---|---|---|---|---|---|
| Training | Arch. | Data | Training | Variant | Architecture | Data | | Training | Architecture | Data | |
| Img. Class. | ViT-B (86M) | ImageNet | MoCoV3 | J | ViT-B (86M) | ImageNet | | CLIP | ViT-B (86M) | CLIP | |
| SAM2 | Hiera B+ (69M) | SA-1B | MAE | R | ViT-B (86M) | ImageNet | | MetaCLIP | ViT-B (86M) | CC-400M | |
| | | | Hiera | R | Hiera B+ (69M) | ImageNet | | SigLIP | ViT-B (86M) | Webli | |
| | | | DINO | J | ViT-B (86M) | ImageNet | | SigLIP (SO) | ViT (414M) | Webli | |
| | | | DINOv2 | J & R | ViT-B (86M) | LVD-142M | | SigLIP 512 | ViT-B (86M) | Webli | |
| | | | DINOv2 | J & R | ViT-L (304M) | LVD-142M | | Aim2 | own (300M) | many | |
| | | | | | | | | ViTamin-L2 | own (333M) | DataComp-1 | |

Table 2: Models, their backbone architectures (with parameters) and their training datasets. Supervised means that human-provided labels were used. J and R denote joint-embedding and reconstruction self-supervised learning methods.

The attention computes interactions between all backbone features $\mathbf{F}(\mathbf{x})$ and all query locations, therefore $\mathbf{M}(\sigma)$ has a size of $(H_Q W_Q, HW)$. Each element of $M_{ij}$ depends on the squared Euclidean distance $d_{ij}^2$ between the $i$-th location in the query and the $j$ location in the feature volume and the region size $\sigma$ through the function $M_{ij} = \frac{1}{\sigma\sqrt{2\pi}} \exp\left(-\frac{d_{ij}^2}{2\sigma^2}\right)$. The learnable parameter $\sigma$ is a scalar in each attention head that adjusts the size of the region in the feature volume (locality) which each query interacts with. The output is obtained by a concatenation of all H head outputs $\mathbf{T} = [\mathbf{T}^{(0)}, ...\mathbf{T}^{(H-1)}]$. Similar to the transformer layer, after this layer, each token $\mathbf{T}_i$ is processed independently by a 2-layer multi-layer perceptron (or feedforward layer) FFN. In the hidden layer of FFN, the vector dimension is expanded by a factor of two.

Instead of using a separate query for each output pixel, regions of size $8 \times 8$ are processed jointly for efficiency reasons, i. e. each query $Q_{ij}$ generates 64 pixels of the output. This is realized by re-arranging the FFN output into spatial shape $(H_Q, W_Q, D)$, denoted by $\mathbf{T}'$, and processing it with a small CNN $\gamma$ which yields the final dense output $\gamma(\mathbf{T}')$. The CNN $\gamma$ has three blocks, each composed of convolution and transposed convolution and ReLU non-linearity. These blocks are followed by a final convolution layer, a skip-connection enables efficient learning. The number of channels in the CNN $\gamma$ is given by $D_{\mathrm{CNN}} = \max(D/4, D_{\mathrm{out}})$, with the number of output channels $D_{\mathrm{out}}$ being task-dependent. Dense attentive probing consists of only a single masked cross attention layer.

Since the queries only receive a position as input and can attend to all features, the masked cross-attention, architecture resembles implicit (or coordinate-based) networks (Mescheder et al., 2019; Park et al., 2019), especially PiFU (Saito et al., 2019) and PixelNerf (Yu et al., 2021) which combine implicit networks with feature volumes. The modification of the attention through a bias term is similar to GraphDINO (Weis et al., 2023), ALiBi (Press et al., 2022) and the work of Shaw et al. (2018). In contrast to these approaches, our bias term depends on 2D distances and has a learnable size parameter.

## 4 Experiments

### 4.1 Choice of models and training datasets

We select a broad range of feature backbones that encompass different training paradigms and were trained on different datasets (Tab. 2). This enables us to conduct controlled comparisons along several axes, for instance, datasets and pretraining task. In general, we differentiate between three broad classes of methods: supervised (Dosovitskiy et al., 2021; Ravi et al., 2024), self-supervised (Caron et al., 2021; He et al., 2022; Ryali et al., 2023), and vision-language (Radford et al., 2021; Xu et al., 2024; Fini et al., 2024; Chen et al., 2024b; Liu et al., 2022). The latter involves training on image-caption pairs often obtained from the internet, while self-supervised training operates on images only. Many models are trained on the ImageNet dataset (Russakovsky et al., 2014), but there are several exceptions: All SigLIP models are trained on the Webli dataset, a Google-internal dataset of 10 billion images with 12 billion multi-lingual text-image pairs. MetaCLIP uses a selection of the open LAION dataset (Schuhmann et al., 2021), CLIP is trained on the unpublished CLIP dataset by OpenAI. DINOv2 (Oquab et al., 2023) is trained on the LVD-142M, a Meta-internal dataset of 142M images which were deduplicated and curated to be similar to ImageNet-22k images. In our experiments, we use the DinoV2 version with registers Darcet et al. (2024). The data mix of Aim2

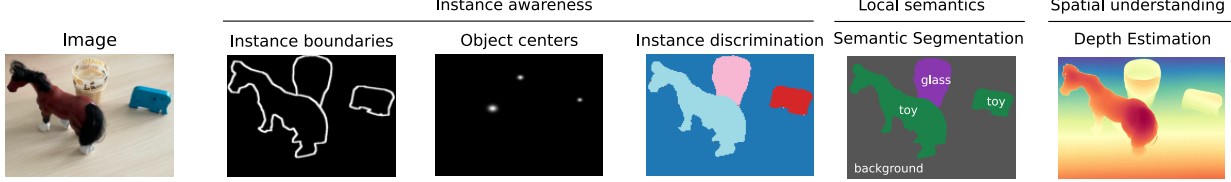

Figure 3: Task overview. We use three tasks to probe instance awareness: object detection based on CenterNet, instance boundaries and instance discrimination. Additionally, we assess local semantics and spatial understanding.

(Fini et al., 2024) contains DFN-2B (Fang et al., 2024a), COYO (Byeon et al., 2022), the proprietary HQITP dataset and synthetic data.

We decide to mainly focus on vision transformers as many approaches share the same architecture and checkpoints are available for a large number of training paradigms. To ensure comparability with other work and control for model architecture, we primarily use ViT-B/16 and similarly sized models in our experiments. We also include larger models in some cases to obtain an estimate of how much performance can be improved simply by scaling-up model size. Pretrained-weights are obtained from the timm package (Wightman, 2019) or the code repositories of the methods.

## 4.2 Experiment design

We provide images in the native resolution of the respective backbones to prevent out-of-distribution input. For a fair comparison, we use images of size $224 \times 224$ as the basis for all tasks, if not indicated otherwise. When backbones require a higher native input resolution, we scale the $224 \times 224$ image to the respective resolution. For generating predictions, we make use of the capability of our model to decouple input and output resolution (see Sec. 3): The output size is fixed to $224 \times 224$ for all models (even for backbones with larger input image sizes), ensuring a fair comparison across models and limiting the advantage of large input sizes for the backbone. In the masked cross attention, we use a dimension $D = 16$ and eight attention heads (i.e. two dimensions per head). Although the readout could attend to backbone activations at different layers we opt for using only the last layer, motivated by Li et al. (2022) who found that a simple feature pyramid using the last layer without top-down connections works best.

We pursue a straightforward approach to comparison: We freeze the backbone features, train the small readout in a supervised way and evaluate on a held-out test set. The rationale is that the low expressivity and capacity of the readout forces it to directly rely on the feature volume for making a dense prediction. This setup differs from conventional task heads (e.g. in object detection) which are able to perform more complex computations on the features. For example, using Faster R-CNN on top of a ResNet50 backbone adds around 18 million parameters to the model.[1]

## 4.3 Evaluation tasks

To characterize a broad spectrum of traits of the backbones we implement three task categories: probing instance awareness, local semantics and spatial understanding (Fig. 3).

**Instance awareness.** In this task we evaluate how well the features are able to disentangle individual instances, for example multiple apples in a bowl. In the field of object-centric machine learning (Burgess et al., 2019) models are designed to disentangle instances. Here we assess the extent to which instance discrimination is already encoded in the features of different backbones as a result of their pretraining. We consider the following three notions of how instances can be encoded:

---

[1]These calculations were obtained using the Faster R-CNN implementation in PyTorch vision (Paszke et al., 2019), in more detail FPN: 3.3M, RPN: 0.6M, ROI heads: 14.3M parameters.

- Instance boundaries: The objective is to outline individual objects in the image. We frame this problem as a binary segmentation and consequently use an output dimension $D_{\text{out}} = 1$ as well as the binary cross-entropy loss function on the output.

- Object centers (CenterNet heatmaps): CenterNet (Zhou et al., 2019) is a detector that uses class-wise heatmaps for detection. We ignore classification and use a single center map. In a second heatmap, bounding box sizes are regressed. Both maps are predicted using individual dense attentive probing readouts with $D_{\text{out}}^{\text{center}} = 1$ and $D_{\text{out}}^{\text{size}} = 2$. As proposed in the original paper, we obtain detections by finding 9-neighborhood maxima in the center heatmap and extracting the bounding box sizes at these locations. We further process the detections with non-maximum suppression. For comparison we report average precision of large objects due to the small (in particular for detection) input image size of $224 \times 224$.

- Instance discrimination: Another way to encode instances is to generate a latent space where features within instances are the same (or similar) while being different to all other instances. If this works perfectly, clustering the latent vectors of all pixels would yield instances. This task is sometimes also called coloring (Novotny et al., 2018). We train on only 8,000 sample images and treat every instance as an individual class (resulting in around 60,000 classes). We first use dense attentive probing to map the features to a latent space (in our case $D_{\text{out}} = 32$). Then, a linear layer maps each local 32-dimensional feature to a probability over all instances in the dataset. Thus, the problem is essentially framed as semantic segmentation with 60,000 classes. This way, the latent features before the classification head learn to discriminate instances. For testing, we cluster these features obtained from unseen images. For clustering we use k-means and provide the ground-truth number of instances as well as a foreground mask. Then we compare the predicted foreground instances with the ground truth instance segmentation based on the adjusted rand index.

For these experiments, we use the COCO dataset (Lin et al., 2014), with the 5,000 images from the validation set being used for testing. For the instance discrimination task, we compute the ARI (adjusted rand index) test scores only on images with at least three large objects (resulting in a subset of 754 images). Note, these tasks do not involve classifying the instances into object categories, unlike typically done in instance segmentation (this is assessed below in "local semantics").

**Local semantics.** A natural choice for evaluating local semantics is a semantic segmentation task. Here we rely on two benchmarks: Pascal VOC 2012 (Everingham et al., 2015) and COCO Stuff (Caesar et al., 2018). The Pascal VOC 2012 encompasses a fairly small set of only 1,464 training images. For COCO Stuff, we train on 100,000 images. We account for the larger number of classes in COCO Stuff by setting the internal dimension of the CNN, $D_{\text{CNN}}$, to 32.

**Spatial understanding.** To assess how well the features capture the 3D structure of the scene, we implement the well-known monocular depth estimation task: The models need to infer the depth (i.e. position along the z-axis) for every pixel of the visible scene based on the features provided by the backbone. We frame this as a depth map estimation problem, i.e. $D_{\text{out}} = 1$, relying on the NYUv2 dataset (Nathan Silberman & Fergus, 2012) for training and testing the depth estimation readout. We first scale the input images to a resolution of $216 \times 288$ and then to the native resolution of the backbones.

## 5 Results

We begin this section by presenting our main result, the comparison of backbone performance and conclusions we draw from it. We also show that the results are in accordance to existing findings. Then we validate the design of our method through an ablation and comparison to similar baselines. Lastly, we investigate the speed-performance tradeoff for practitioners.

| Cat. | Backbone | $I$ | $F$ | $P$ | Inst. Disc. ARI ↑ | $P_{\text{learn}}$ | Boundaries CE ↓ | IoU ↑ | $P_{\text{learn}}$ | Object Detection $AP_{\text{lg}}$ ↑ | $P_{\text{learn}}$ |
|---|---|---|---|---|---|---|---|---|---|---|---|
| ○ | Random (untrained) | 224 | 14 | 85.8 | 23.5 | 0.053 | 0.2544 | 0.8 | 0.043 | 0.0015 | 0.087 |
| ● | ImageNet | 224 | 14 | 85.8 | 36.3 | 0.053 | 0.1787 | 16.0 | 0.043 | 0.1126 | 0.087 |
| ● | SAM V2 B+ | 1024 | 64 | 80.8 | **50.0** | 0.028 | 0.1394 | 30.2 | 0.018 | 0.1759 | 0.035 |
| ● | MoCo V3 | 224 | 14 | 85.8 | 41.8 | 0.053 | 0.1613 | 21.4 | 0.043 | 0.2078 | 0.087 |
| ● | Dino | 224 | 28 | 85.9 | 41.5 | 0.053 | 0.1433 | 28.3 | 0.043 | 0.2344 | 0.087 |
| ● | Dino V2 | 518 | 37 | 86.6 | 46.8 | 0.053 | **0.1297** | 32.4 | 0.043 | 0.2666 | 0.087 |
| ● | Dino V2 (ViT-L) | 518 | 37 | 304.4 | 46.4 | 0.066 | 0.1450 | **33.8** | 0.056 | **0.2811** | 0.112 |
| ● | MAE | 224 | 14 | 85.8 | 49.8 | 0.053 | 0.1500 | 25.3 | 0.043 | 0.2662 | 0.087 |
| ● | Hiera B+ | 224 | 7 | 69.1 | 44.8 | 0.060 | 0.1717 | 17.1 | 0.050 | 0.2315 | 0.099 |
| ● | CLIP | 224 | 14 | 85.9 | 39.5 | 0.053 | 0.1665 | 19.0 | 0.043 | 0.1391 | 0.087 |
| ● | CLIP (ViT-L) | 336 | 24 | 303.6 | 40.6 | 0.066 | 0.1543 | 24.3 | 0.056 | 0.2005 | 0.112 |
| ● | MetaCLIP | 224 | 14 | 85.9 | 38.4 | 0.053 | 0.1669 | 19.1 | 0.043 | 0.1667 | 0.087 |
| ● | SigLIP-224 | 224 | 14 | 85.8 | 37.8 | 0.053 | 0.1718 | 17.2 | 0.043 | 0.1677 | 0.087 |
| ● | SigLIP-384 | 384 | 24 | 86.1 | 39.1 | 0.053 | 0.1564 | 23.3 | 0.043 | 0.1836 | 0.087 |
| ● | SigLIP-512 | 512 | 32 | 86.5 | 38.9 | 0.053 | 0.1496 | 26.2 | 0.043 | 0.1910 | 0.087 |
| ● | SigLIP-SO | 512 | 36 | 413.7 | 40.7 | 0.073 | 0.1496 | 25.7 | 0.062 | 0.1952 | 0.125 |
| ● | Aim2 | 336 | 24 | 309.6 | 40.0 | 0.066 | 0.1556 | 23.5 | 0.056 | 0.2011 | 0.112 |
| ● | ViTamin | 384 | 24 | 333.0 | 42.1 | 0.066 | 0.1535 | 24.4 | 0.056 | 0.1648 | 0.112 |

Table 3: Instance awareness results in all three categories. $I$: image size. $F$: Size of feature volume. $P$ and $P_{\text{learn}}$: Number of all and learnable parameters, respectively, in millions. Metrics: Adjusted rand index (ARI), cross-entropy (CE), intersection over union (IoU), average precision for large objects ($AP_{\text{lg}}$).

| 0 | Backbone | I | F | P | Pascal VOC2012 CE | IoU | $P_{learn}$ | COCO Stuff CE | IoU | $P_{learn}$ | Depth RMSE | $P_{learn}$ |
|---|---|---|---|---|---|---|---|---|---|---|---|---|
| ○ | Random (untrained) | 224 | 14 | 85.8 | 2.7364 | 3.5 | 0.047 | 5.0889 | 0.1 | 0.088 | 1.757 | 0.043 |
| ● | ImageNet | 224 | 14 | 85.8 | 0.3924 | 61.0 | 0.047 | 1.5140 | 28.1 | 0.088 | 0.713 | 0.043 |
| ● | SAM V2 B+ | 1024 | 64 | 80.8 | 0.6005 | 33.9 | 0.022 | 2.0630 | 12.6 | 0.062 | 0.714 | 0.018 |
| ● | MoCo V3 | 224 | 14 | 85.8 | 0.3274 | 63.3 | 0.047 | 1.4417 | 27.3 | 0.088 | 0.606 | 0.043 |
| ● | Dino | 224 | 28 | 85.9 | 0.3397 | 63.8 | 0.047 | 1.3423 | 30.2 | 0.088 | 0.585 | 0.043 |
| ● | Dino V2 | 518 | 37 | 86.6 | 0.1259 | 83.4 | 0.047 | 1.0468 | 42.8 | 0.088 | 0.425 | 0.043 |
| ● | Dino V2 (ViT-L) | 518 | 37 | 304.4 | 0.1173 | **85.3** | 0.060 | **1.0347** | **43.8** | 0.101 | **0.398** | 0.056 |
| ● | MAE | 224 | 14 | 85.8 | 0.3471 | 60.2 | 0.047 | 1.4822 | 25.3 | 0.088 | 0.580 | 0.043 |
| ● | Hiera B+ | 224 | 7 | 69.1 | 0.3344 | 61.5 | 0.054 | 1.5007 | 25.7 | 0.094 | 0.523 | 0.050 |
| ● | CLIP | 224 | 14 | 85.9 | 0.2710 | 68.3 | 0.047 | 1.3367 | 32.0 | 0.088 | 0.603 | 0.043 |
| ● | CLIP (ViT-L) | 336 | 24 | 303.6 | 0.2228 | 74.9 | 0.060 | 1.2623 | 35.5 | 0.101 | 0.515 | 0.056 |
| ● | MetaCLIP | 224 | 14 | 85.9 | 0.2580 | 69.4 | 0.047 | 1.3427 | 31.9 | 0.088 | 0.576 | 0.043 |
| ● | SigLIP-224 | 224 | 14 | 85.8 | 0.2898 | 67.1 | 0.047 | 1.3355 | 33.0 | 0.088 | 0.598 | 0.043 |
| ● | SigLIP-384 | 384 | 24 | 86.1 | 0.2137 | 73.9 | 0.047 | 1.2565 | 35.4 | 0.088 | 0.568 | 0.043 |
| ● | SigLIP-512 | 512 | 32 | 86.5 | 0.2203 | 75.7 | 0.047 | 1.2435 | 36.3 | 0.088 | 0.565 | 0.043 |
| ● | SigLIP-SO | 512 | 36 | 413.7 | 0.1908 | 77.9 | 0.066 | 1.1737 | 38.8 | 0.107 | 0.488 | 0.062 |
| ● | Aim2 | 336 | 24 | 309.6 | 0.2244 | 75.2 | 0.060 | 1.1957 | 37.5 | 0.101 | 0.498 | 0.056 |
| ● | ViTamin | 384 | 24 | 333.0 | 0.1843 | 77.9 | 0.060 | 1.1757 | 37.9 | 0.101 | 0.492 | 0.056 |

Table 4: Local semantics results on Pascal and COCO Stuff as well as spatial understanding on NYUv2 (right). $I$: image size. $F$: Size of feature volume. $P$ and $P_{\text{learn}}$: Number of all and learnable parameters, respectively, in millions. Metrics: cross-entropy (CE), intersection over union (IoU), mean-squared error (MSE)

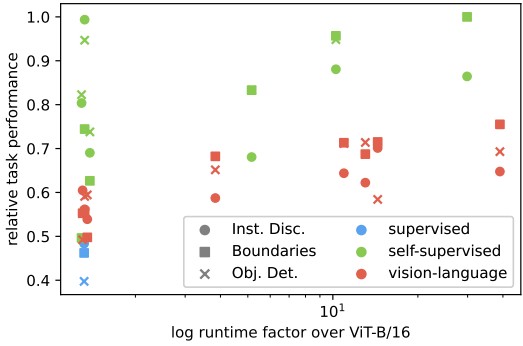 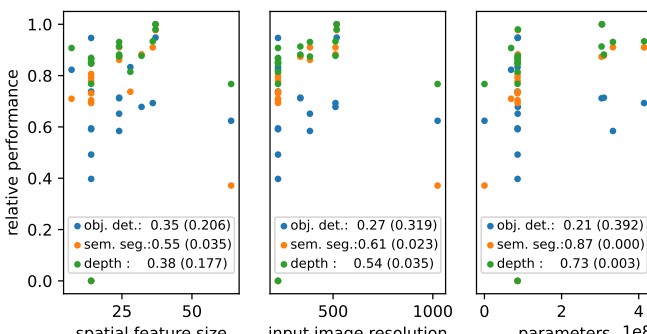

(a) Self-supervised methods consistently outperform vision-language methods across all three instance awareness tasks.

(b) Spearman rank correlation of relative performance to spatial feature size, input image resolution and number of parameters. P-values are shown in brackets (with multiple comparison correction using the Benjamini-Hochberg method).

Figure 4: Comparison of relative task performance. To show multiple tasks in a single figure, we report the relative task performance, for which the original scores were mapped linearly into the interval [0,1].

## 5.1 Comparison of backbone performance

A main goal of our work is to provide a fair comparison of different backbones regarding instance discrimination, local semantics and spatial understanding. We aim to highlight strengths and weaknesses of each backbone. Results (Tab. 3) indicate that DINOv2 has the best instance awareness. This finding is consistent with recent work on whole-image classification (Oquab et al., 2023) and other tasks such as physics (Zhan et al., 2024) and 3D understanding (El Banani et al., 2024; Bonnen et al., 2024). The backbone of SAM2 does not outperform other backbones considerably, this is surprising given that it was trained on instance discrimination. It suggests SAM2's mask decoder (which we did not use here) is a crucial component. Interestingly, self-supervised methods consistently outperform vision-language models in instance awareness (Fig. 4a). Despite following the same reconstruction-based training, MAE performs substantially better than Hiera in instance awareness while the opposite is true for semantic segmentation and depth estimation. Among the vision-language models CLIP, MetaCLIP and SigLIP we did not find meaningful differences. We investigated the effect of backbone spatial feature size, backbone input resolution (images had a constant resolution and were upscaled to native backbone resolution for fairness) and number of parameters of the backbone by considering the Spearman rank correlation (Fig. 4b). While we do not find significant results for object detection, both semantic segmentation and depth benefit from larger feature volumes, better input resolution and more backbone parameters. Especially, semantic segmentation and depth show a strong correlation of 0.87 and -0.79 to the number of backbone parameters.

Vision-language models tend to show stronger local semantics (Tab. 4). While all vision-language models with 224px input size show similar performance, the larger versions of SigLIP (i.e. 384px and the shape optimized SO version) perform better but at a higher cost. Also in this evaluation, DINOv2 achieves the best scores. All things considered, possibly the most striking finding is the dominance of DINOv2. While one might argue that this is due to large image sizes and feature volumes, the mediocre performance of SigLIP-512, Hiera-B+ and (partly) SAM V2 show that it cannot be the only factor.

The evaluation on spatial understanding shows mixed results. Larger backbones tend to perform better, with the exception of Hiera-B+. Again, DINOv2 performs best, in this case by a large margin.

## 5.2 DeAP performance correlates with performance of object-centric methods and related benchmark

Next, we relate our findings to previous work. Specifically, we consider previous work in object-centric representation learning and a classification-based evaluation (Fig. 5). Object-centric learning shares the goal of disentangling instances but achieves this through specific model architectures, whereas we evaluate

Figure 5: We compare our results with reported scores on two object-centric learning methods (three leftmost panels) (Aydemir et al., 2023; Seitzer et al., 2023)), fine-tuned Faster R-CNN backbones (Goldblum et al., 2024) and CV-Bench 2D and 3D (Tong et al., 2024).

model-agnostic features for instance-specific signals. In these experiments we compare how well the object-centric scores match our instance discrimination scores for the same backbones. Relating to the instance clustering foreground adjusted rand-index, which indicates how well instances are disentangled, by Aydemir et al. (2023) we find an almost linear relationship between their and our instance discrimination scores. Also compared DINOSAUR (Seitzer et al., 2023), we find an almost linear relationship over three backbones (Dino ViT-S, Dino ViT-B and MAE).

The evaluation of Goldblum et al. (2024) shares the goal of characterizing current backbones with our work but puts more emphasis on out-of-distribution and backbone architecture. While a positive relation between their and our scores is recognizable, the trend is less pronounced than in previous experiments. CV-Bench-2D and CV-Bench-3D (Tong et al., 2024) were recently introduced to assess the capabilities of multi-modal language models and provide scores for several vision backbones. While the variance is larger, we can see the same trends again: For the semantic score of CV-Bench-2D there is a positive correlation with our semantic segmentation measurements while there is a negative correlation between the spatial scores of CV-Bench-3D and our monocular depth prediction errors (where lower is better). In summary, our method can be used to obtain similar insights on relative backbone performance as more expensive and slower evaluation methods.

### 5.3 Ablation on readout design

We next explore design choices of our dense attentive probing readout by varying relevant hyperparameters (Tab. 5). The internal dimension of the method correlates positively with performance: while using 24 dimensions instead of 16 (base) improves performance but also requires 50 pp more parameters. An analogous observation can be made for the 8 dimensional version. The introduction of the $\sigma$ parameter and its adaptivity per head is crucial for good performance as shown by the performance drop in versions where no masking is used ("no masking", i.e. attention between all locations is possible) and a learnable $\sigma$ is shared for all heads ("no indiv. $\sigma$"). This suggests that information is organized at different levels of locality in the feature volumes. Ignoring all query information and thus attending only based on masks ("only-mask") works well in some cases, in particular for the MAE backbone. The approach seems robust with respect to the number of heads, as the 4-head version performs similarly. The relevance of the feedforward network (FFN) varies across backbones and tasks in some cases quite strongly. The CNN ($\gamma$) accounts for small performance improvements but can be seen as a non-critical component.

### 5.4 Comparison to dense readout baselines

Next, we ask whether alternative architectures with a similar parameter budget could perform competitively to dense attentive probing. To investigate this question, we implement three dense readouts capable of obtaining a high resolution prediction based on a low-resolution feature volume $\mathbf{F}(\mathbf{x})$ of size $(HW, D)$. Specifically, we compare against the following baselines:

- **Bicubic Interpolation (Bicubic):** A natural choice to increase resolution is to interpolate in the feature volume $F$. This baseline replaces only the cross-attention of our model by a projection to a low dimension followed by a non-learnable bicubic interpolation operation. The rest of the baseline,

| names | rel. Params. | Object Detection: $AP_{lg}$ ↑ | | | Semantic Segmentation: IoU ↑ | | | Depth: RMSE ↓ | | |
|---|---|---|---|---|---|---|---|---|---|---|
| | | Dino V2 | SigLIP-384 | MAE | Dino V2 | SigLIP-384 | MAE | Dino V2 | SigLIP-384 | MAE |
| base | 100% | 0.2666 | 0.1836 | 0.2662 | 83.4 | 73.9 | 60.2 | 0.4248 | **0.5684** | 0.5804 |
| 8-dim | 51% | 0.2448 | 0.1722 | 0.2529 | 82.8 | 70.4 | 55.6 | 0.4279 | 0.6041 | 0.5592 |
| 24-dim | 154% | **0.3150** | **0.2103** | **0.2871** | 84.0 | **75.3** | 60.2 | **0.4161** | 0.5746 | **0.5453** |
| 4 heads | 100% | 0.3027 | 0.1971 | 0.2633 | **84.9** | 74.2 | 59.4 | 0.4283 | 0.5894 | 0.5536 |
| No masking | 100% | 0.1791 | 0.1196 | 0.1712 | 50.5 | 38.1 | **35.1** | 0.5992 | 0.7132 | 0.6801 |
| No indiv. $\sigma$ | 100% | 0.2409 | 0.1520 | 0.2429 | 83.8 | 70.2 | 58.6 | 0.4670 | 0.6070 | 0.5864 |
| No FFN | 94% | 0.2409 | 0.0277 | 0.1234 | 83.7 | 73.2 | 56.4 | 0.4429 | 0.5829 | 0.5650 |
| No CNN $\gamma$ | 97% | 0.2969 | 0.1876 | 0.2632 | 83.2 | 73.4 | 59.7 | 0.4269 | 0.5711 | 0.5527 |

Table 5: Ablation of model parameters and components. The base model is the variant we use in all other experiments. We vary our model's internal dimension (-dim), number of attention heads, use of masking, use of individual $\sigma$, and removal of FFN or CNN $\gamma$.

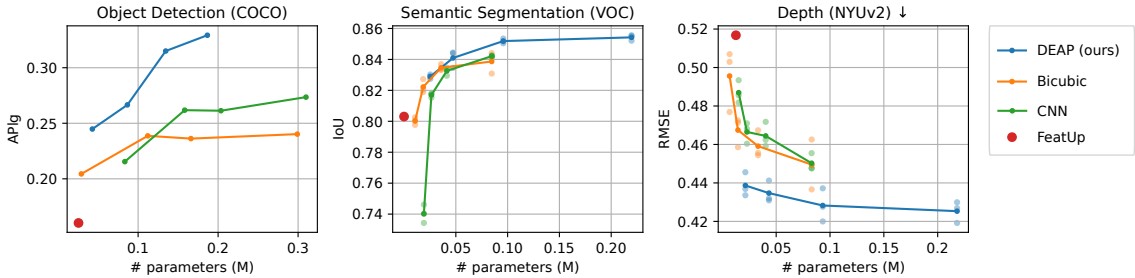

Figure 6: Dense attentive probing outperforms CNN, interpolation and FeatUp-based readouts on DinoV2 features (instances: ↑ = better; semantic segmentation ↑; depth ↓). For semantic segmentation and depth estimation the lines represent averages over three runs. For the largest bicubic run on object detection we report the best score of three runs due to training instability.

    i.e. the MLP and CNN, is identical to dense attentive pooling to isolate the effect of replacing the cross-attention.

- **Transposed Convolutions (CNN):** A common component in many dense prediction architectures are transposed convolutions. This operation reverses downsampling by applying learned filters that generate spatially larger outputs. Analogous to the bicubic interpolation baseline, we replace the masked cross-attention with a single transposed convolution layer, while we use the same CNN for upsampling. A disadvantage of this baseline is that the output resolution depends on the feature volume resolution, e.g. increasing the feature volume from by a factor of two would also increase the output size by a factor of two.

- **FeatUp:** Recently, the FeatUp (Fu et al., 2024) method was introduced that up-scales a feature volume under consideration (conditional on) of the input image. We employ this image-aware upsampling technique to up-scale the feature volumes $\mathbf{F}(\mathbf{x})$ and add a linear projection that maps to the task-defined output space.

We find dense attentive probing to be more parameter efficient and to achieve better scores than the baselines across all three tasks (Fig. 6). An additional advantage of our method over CNNs is decoupling input and output resolution, in a CNN, a larger feature volume size would result in a larger output. FeatUp is highly parameter efficient but has high memory demands and requires long computation times (factor 4 compared to dense attentive probing).

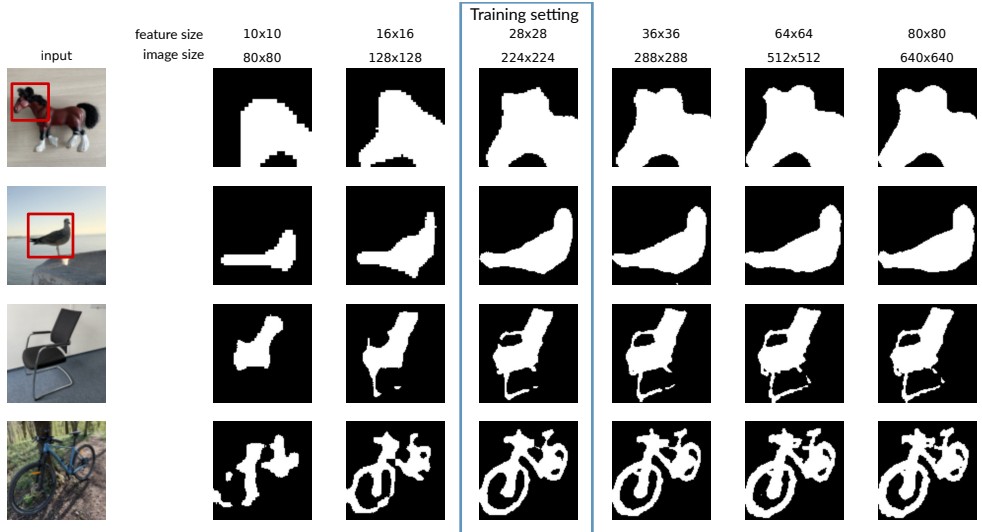

Figure 7: After training, our readout can be queried to output different resolutions from the same backbone. Here we use a DINOv2 backbone trained on Pascal VOC.

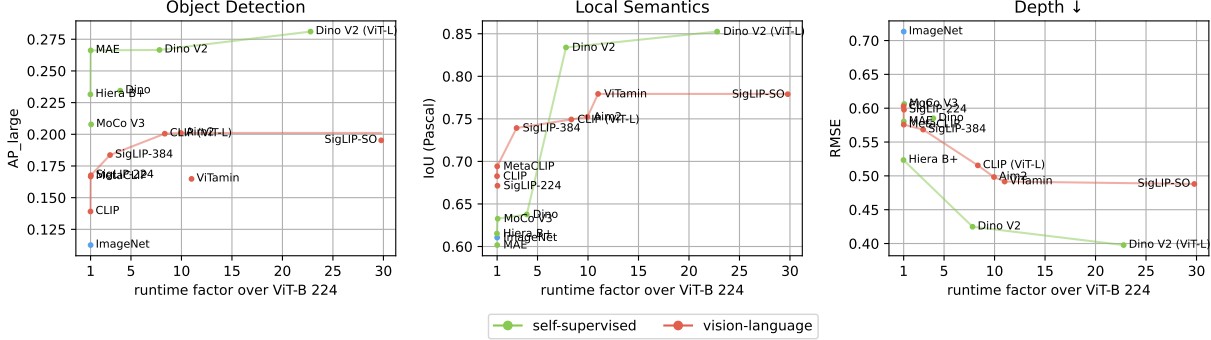

Figure 8: Inference speed over performance on three tasks, relative to the ViT-B/16 with 224px image input (the fastest and most frequent architecture in our evaluation). The lines depict the pareto fronts.

## 5.5 Variable output size

To ensure a fair comparison, the readout size is fixed in the previous experiments. However, it is possible to generate outputs at an arbitrary resolution, because the adapter takes positions as inputs (similar to implicit neural fields). Instead of using the standard query position grid for a 224 px output, we can sample different query coordinates at test time (Fig. 7). The results show that our method generalizes well to higher resolutions, with some details being resolved better at higher resolutions (see for example the head of the horse and the chair).

## 5.6 Speed-performance tradeoff

For practitioners, runtime limitations can constrain the selection of viable backbones, for example when a minimal latency is required. Therefore, we explore the relationship between task performance and inference speed in more detail (Fig. 8). We measure the time to run ten batches with eight samples each in inference mode (i.e. without gradient computation). The fastest model in our evaluation set is the ViT-B/16 at a resolution of 224 pixels. Therefore, we indicate the factor by which the runtime is extended with respect to this model. For example, the slowest model, SigLIP-SO takes around 30 times as long as the ViT-B/16 reference model. The results show that, depending on the task, there are fast models that achieve good

performance, namely MAE for object detection and Hiera B+ for depth prediction. For local semantics we find a stronger dependency between model sizes and performance, i.e. larger models are required for good performance.

# 6 Conclusion

In this work we proposed dense attentive probing, a fast and parameter-efficient readout for evaluating the representational expressiveness of trained backbones on dense prediction. For example, our standard training for a readout on a ViT-B/16 224 pixel backbone on Pascal VOC adds less than 70,000 parameters and trains in less than 16 minutes (using a single Nvidia RXT2080 GPU). We used dense attentive probing to systematically analyze common vision backbones with respect to the three complementary aspects: instance awareness, depth and local semantics. Our results suggest that the backbones of the DINOv2 family are highly capable: It is the best ViT-B model across all experiments. This finding is in line with prior research on complementary tasks (Oquab et al., 2023; Zhan et al., 2024; El Banani et al., 2024; Bonnen et al., 2024). Using DINOv2 with a ViT-L backbone improves performance further but at the cost of a three times longer runtime. Classic supervised image classification pretraining on ImageNet results in inferior performance compared to the self-supervised and vision-language paradigms. Excluding DINOv2, we identified the trend that local semantics is better captured by language-vision models while reconstruction-based self-supervised learning leads to features with better instance awareness. This indicates that combining both paradigms could be a promising direction for future research.

For practitioners, DINOv2 is a natural choice if enough compute is available. For compute-constrained cases, the decision is more complex: MAE is recommendable for instance related tasks while CLIP-based models (e.g. MetaCLIP) show good local semantics. For spatial understanding Hiera-B represents a good trade-off between performance and inference speed. We plan to retain an online leaderboard where new backbones can easily be incorporated to help tracking future progress of dense prediction performance.

# 7 Limitations

While we use a fairly small readout (in terms of parameters) that can adapt to multiple locality scales in the features through the learnable masks, even this readout has inductive biases and can favor certain backbones such that results might get distorted. We deliberately opted for a small readout to directly measure the feature performance. Consequently, this limits the degree to which the features can be recombined and processed. In fact, this could be the reason why the SAM2 backbone performs comparably poorly in instance awareness in our hands: it does rely on a relatively "heavy" decoder. A more direct comparison to object-centric approaches would be desirable, but is challenging as these approaches explicitly encode objects (e. g. in attention slots) which can be compared to ground truth.

The current selection of tasks we evaluated is limited to three broad categories and a few instances of those. Adding additional task categories (e. g. as in Taskonomy (Zamir et al., 2018)) would be desirable in the future for a broader characterization of backbones.

# 8 Acknowledgements

This publication was funded by the Deutsche Forschungsgemeinschaft (DFG, German Research Foundation) - Project-ID 454648639 - SFB 1528

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

# A    Appendix

## A.1    Implementation

We use the Adam optimizer with a learning rate of 0.001, except for boundary prediction and depth where it is set to 0.002. We use 8 attention heads in all models. On COCO and Pascal we use the validation sets for testing, while model selection is carried out on a separate part of the training set via validation loss.

| name | BS | LR | WD | iterations | val intv. | img size | heads | dim | base size |
|---|---|---|---|---|---|---|---|---|---|
| Pascal VOC2012 | 32 | 0.001 | 0.010 | 6000 | 250 | 224 | 8 | 16 | 28 |
| COCO Stuff | 32 | 0.001 | 0.010 | 20000 | 250 | 224 | 8 | 16 | 28 |
| NYUv2 Depth | 32 | 0.001 | 0.010 | 3000 | 100 | [216, 288] | 8 | 16 | 28 |
| Instance Discrimination | 32 | 0.001 | 0.010 | 20000 | -1 | 224 | 8 | 16 | 28 |
| Boundariers | 32 | 0.001 | 0.010 | 10000 | 250 | 448 | 8 | 16 | 56 |
| CenterNet | 32 | 0.001 | 0.010 | 20000 | 1000 | 448 | 8 | 16 | 56 |

BS,LR and WD correspond to batch size, learning rate and weight decay respectively.

## A.2    Comparison in higher resolution

Some of the evaluated backbones support higher native resolutions. Here we increase the provided resolution to $448 \times 448$ and $432 \times 576$ for semantic segmentation and depth respectively. We find small improvements in performance for SigLIP-512 and DinoV2 due to the increased resolution.

| Model | Inp. size | NYUv2 Depth RMSE | Pascal VOC2012 IoU |
|---|---|---|---|
| **Original resolution** | | | |
| SigLIP-512 | 512 | 0.5648 | 75.7 |
| ViTamin | 384 | 0.4916 | 77.9 |
| Dino V2 | 518 | 0.4248 | 83.4 |
| **Higher resolution** | | | |
| SigLIP-512 | 512 | 0.5571 | 78.0 |
| ViTamin | 384 | 0.4945 | 77.2 |
| Dino V2 | 518 | 0.4208 | 85.0 |

## A.3    Inference Speed

In Fig. 9 we show the inference speeds relative to the fastest model (ViT-B/16).

## A.4    Feature Visualization

We visualize the backbones output features (Fig. 10) by reducing the number of feature dimensions to three and interpreting these three dimensions as RGB color.

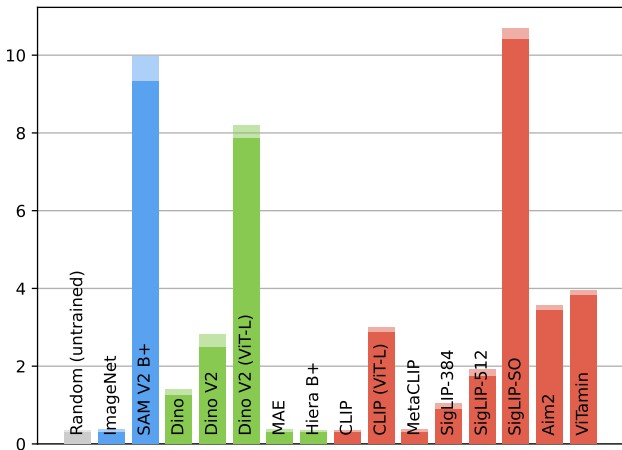

Figure 9: Inference speed of selected methods. Light bars on the top represent runtime of the dense attentive probing.

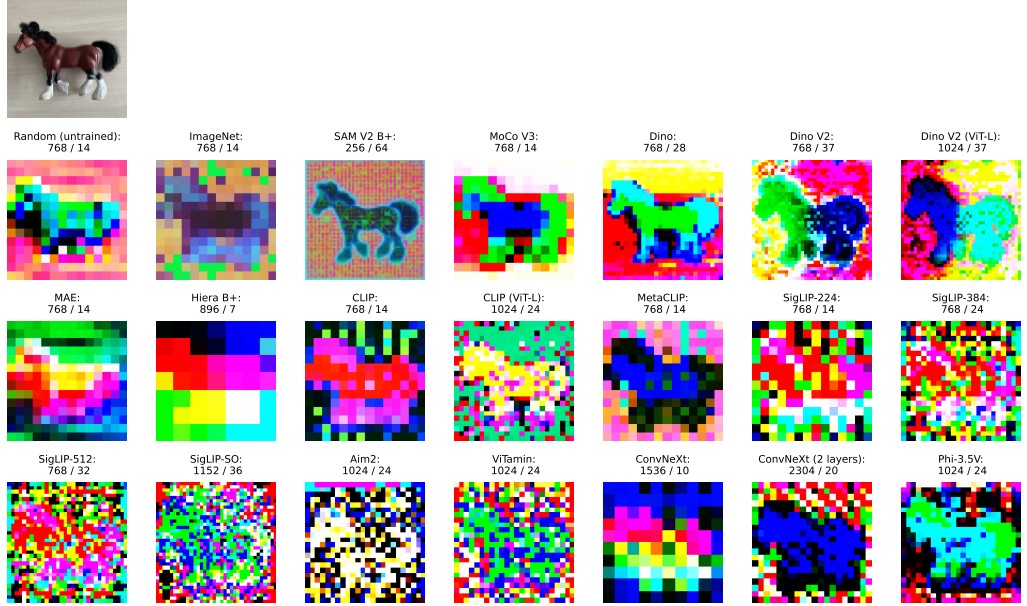

Figure 10: PCA Visualization of the backbone features. The numbers below the backbone names denote feature dimension / spatial size of the feature volume.

