# OpenReview forum: "Characterizing Vision Backbones for Dense Prediction with Dense Attentive Probing"
_TMLR — Accepted by TMLR_

### Review · Reviewer_AftN · 2025-04-22

**Summary Of Contributions:**

The paper proposes an architecture and methodology for probing vision models for dense prediction tasks consisting of 1) an dense attentive probe (DAP) and 2) a set of dense prediction task.  The dense attentive probe takes a grid of positional queries, applies attention with relative position bias to features from the vison backbone, and the upsamples that using a CNN to obtain a final dense readout.  The DAP is trained on a set of dense prediction tasks consisting of 1) instance awareness tasks (boundary detection, center detection, and instance discrimination on MS COCO), 2) local semantics - semantic segmentation (on Pascal VOC 2012 and COCO stuff), and 3) spatial understanding - monocular depth understanding on NYUv2.  Experiments are conducted to compare different pre-trained ViT-based vision models covering a range of training objectives (e.g. MAE, DINO, SigLIP, etc).

The main contributions are 1) architecture for probing vision models for dense prediction, 2) set of tasks for training the dense attentive probe, and 3) set of experiments comparing different pretrained models.

**Audience:**

Yes

**Claims And Evidence:**

Yes

**Requested Changes:**

The reviewer believe that considerable improvements to writing and presentation is needed before the word is ready for publication.

1. The related work need to be considerably improved
- For instance, there is a lot of information about PEFT, but it is not clear how relevant those methods are to this work.
- This reviewer did not found the section on "Feature evaluation" to be that relevant or informative.  Instead, the reviewer expected to see more on alternative methods for dense probing, feature evaluation (e.g. recent efforts in evaluating visual backbones [1,2] ) and potentially work on interpretability (e.g. GradCam [3] for visualizing features).
- It would be good to include a discussion between this work and work on direct image segmentation (e.g why would those studies be not appropriate for dense probing)
2. Table 1 is not referenced
3. Figure 1 is hard to follow and match with the caption
- It's not clear what parts of the figure depict the queries, is the small CNN the CNN gamma, where are the dense readout, Gaussian function, etc.
- The attention part is also not described, even though there is is large parts of figure devoted to it
- Use symbols in caption to connect to figure, and add names of components to the figure
4. Table 2 is hard to understand.  Table needs columns headers.  What does the entry "Supervised" mean in the Supervised table?
5. Figure 2 - why only include the instance understanding tasks?  Why not include the semantic segmentation and monocular depth estimation task?
6. Section 3, when introducing $M(\sigma)$, the key idea should be introduced vs just introducing the notation.  It seems also appropriate to relate this to other work that uses biases with attentino (e.g. ALiBi [4]).
7. The experiment results (section 5) should be improved to have a better overview of the sections and the experiments.  While the tasks and a set of baselines was described in section 4, section 5 was hard to follow as it jumped into the results.  One issue is that section 4.2 (baselines) was introduced and then there was a set of experiments without considering this baselines.  There should be more context provided for each set of experiments (e.g. for section 5.2, why the specific models are selected, what does the different conditions mean, what is being measured, etc.)
8. There should be more discussion of 1) design choices in probing approach (e.g. which parameters should be adjusted for different use cases), and 2) how the findings of this work relates to findings from prior work.

**References**
[1] V-PETL Bench: A Unified Visual Parameter-Efficient Transfer Learning Benchmark [Xin et al. NeurIPS 2024]

[2] Lexicon3D: Probing Visual Encoding Models for Complex 3D Scene Understanding [Man et al. NeurIPS 2024]

[3] Grad-CAM: Visual Explanations from Deep Networks via Gradient-based Localization [Selvaraju et al. CVPR 2017]

[4] Train Short, Test Long: Attention with Linear Biases Enables Input Length Extrapolation [Press et al. ICLR 2022]

**Strengths And Weaknesses:**

**Strengths**
1. Establishing a probing protocol for evaluating pretrained vision models for dense prediction is important and timely
2. There are experiments using the proposed probing approach to evaluate a variety of different vision models
3. The introduction is well-written and compelling

**Weaknesses**
1. The writing and presentation need to be improved considerably
  - There seems to be a mismatch between the related work and the goals specified in the introduction
  - The figures and tables are hard to understand and do not really connect with main paper text
2. Limited discussion of relevant prior work
  - No discussion of simpler efforts on dense probing (e.g. dense probing in DINOv2)
3. Due to the limited discussion of prior work, it was not clear whether this work compared to the appropriate prior methods and whether this work is this better in some way.  It's also not clear whether the selected tasks is better than evauating using more standard instance segmentation or 3D scene understanding for spatial understanding.
4. Due to Table 1 (which is not referenced) and the section devoted to parameter efficient fine-tuning methods, the position of this paper became unclear: is this a paper focused on an adapter method for dense prediction or a probing paper?
5. It's unclear whether the baselines in section 4.2 are true baselines or should they be considered as alternatives to upsampling the readout.
6. Limited discussion of the impact of various design choices.  What parameters are selected just for the experiments in this paper vs which parameters are recommended because of some desirable features?
7. The experiments use images of size 224x224 for all tasks, and for image backbones that have higher native resolution, the images are scaled up.  As the outputs are also at 224x224, this setup does not seem to take advantage of the higher resolution offered by some of the backbones.  Would it be make more sense to separately evaluate backbones that are higher resolution with higher resolution inputs and outputs?
8. The reviewer found the experimental results difficult to follow and the figures and tables hard to interpret
  - For instance, there was too many different factors in Table 4 to see if there were any conclusions as to what at the important factors to having a good pretrained image model (training dataset, pretraining objective, image resolution, etc.).  It was also not clear what the row for ImageNet was - is that the original ViT model with supervised training on ImageNet?  Figure 3 was more useful.

---

> ### Author Response · Authors · 2025-06-12
>
> We thank the reviewer for the thoughtful comments and suggestions. Suggested changes are indicated by **(C#)**
>
>
> ### Presentation
> > Position unclear ... Is this a paper focused on an adapter method for dense prediction or a probing paper?
>
> The focus of the paper is on probing common backbones for dense prediction tasks. We do not seek to outperform state-of-the-art methods but aim to develop a simple method which is well suited to characterize features. We clarified this in the introduction.
>
>
> > Reference Table 1 **(C2)**
>
> We added a reference to table 1 as part of the revision of the related works section.
>
>
>
>
> > Figure 1 is hard to follow ... **(C3)**
> > It's not clear what parts of the figure depict the queries, is the small CNN the CNN gamma, where are the dense readout, Gaussian function, etc.
> > The attention part is also not described, even though there is is large parts of figure devoted to it
> > Use symbols in caption to connect to figure, and add names of components to the figure
>
> We agree that Fig. 1 could be improved and revised figure and caption to incorporate your suggestions.
>
>
>
>
>
>
>
> > Table 2 is hard to understand. Table needs columns headers. What does the entry "Supervised" mean in the Supervised table? **(C4)**
>
> We added heads and clarified that supervised means that the training of the backbones used human-provided labels. We also changed the method name to image classification (instead of supervised) to be more precise.
>
>
> > Figure 2 - why only include the instance understanding tasks? Why not include the semantic segmentation and monocular depth estimation task? **(C5)**
>
> That is a good idea, we revised Fig. 2 accordingly.
>
>
> > Section 3, when introducing M, the key idea should be introduced vs just introducing the notation. **(C6)**
>
> We elaborated the intuition of our approach in the beginning of Section 3 and other bias-based related work, including ALiBi.
>
>
> > there was too many different factors in Table 4 to see if there were any conclusions as to what at the important factors to having a good pretrained image model
>
> Due to the availability of backbones, we cannot conduct controlled experiments for all factors. The results allow to draw some conclusion: DinoV2 performs well across tasks and instances awareness tends to be learned better in self-supervised learning than vision-language pre-training. Based on your comment we conducted further analysis on the relationship between feature volume size, image resolution and backbone parameters to performance. The results are reported in Fig. 3b and the section 5.1.
>
>
>
> ### Experiments
>
>
> > The experiment results (section 5) should be improved to have a better overview of the sections and the experiments. While the tasks and a set of baselines was described in section 4, section 5 was hard to follow as it jumped into the results.  **(C7)**
>
> We added an overview to the results section at the beginning of the section.
> We agree that moving the description of the baselines into the results section improves reading flow.
>
>
> > It's unclear whether the baselines in section 4.2 are true baselines or should they be considered as alternatives to upsampling the readout.
>
> We are unsure if we understand the question correctly.
> The provided baselines are models that fulfill criteria of generating dense output from arbitrary backbones using a small parameter budget, thus enabling a comparison.
> They either part share parts of our model (Bicubic and CNN) or are standalone methods (FeatUp). We clarified the description of the baselines in the paper.
> We also renamed the section to "Comparison to dense readout baselines" to be more precise.
>
>
>
> > Limited discussion of the impact of various design choices. What parameters are selected just for the experiments in this paper vs which parameters are recommended because of some desirable features?
>
> > There should be more discussion of 1) design choices in probing approach **(C8)**
>
> We agree that the design of our model can be ablated in more detail and extended the ablation, including no feedforward network, no CNN and only 4 attention heads.
> We also extended the discussion of the ablation.
>
>
> > Would it be make more sense to separately evaluate backbones that are higher resolution with higher resolution inputs and outputs?
>
> We decided against dividing the comparison into resolution categories in favour of a comparison between a larger set of backbones. As a result, we have to use the smallest compatible resolution of 224x224.
> However, we agree that investigating the effect of resolution can be interesting. Therefore, we conducted a specific experiment for depth and semantic segmentation with high resolution input on three backbones and report results in the appendix.
> Note, we also provide a new analysis on the effect of higher resolution backbones in Fig. 3b.

---

> > ### Author Response · Authors · 2025-06-12
> >
> > ### Related Work
> >
> > > The related work need to be considerably improved **(C1)**
> >
> > > For instance, there is a lot of information about PEFT, but it is not clear how relevant those methods are to this work.
> >
> > We consider PEFT to be relevant as it shares the goal of adapting backbones to new tasks with a small number parameter budget. We motivate this at the beginning of the paragraph.
> >
> >
> > > [...] "Feature evaluation" to be that relevant or informative. Instead, the reviewer expected to see more on alternative methods for dense probing, feature evaluation [...] and potentially work on interpretability [...]
> >
> > We rewrote the respective section and incorporated the suggested papers as well as several other papers on dense probing and feature evaluation into the discussion. We consider interpretability to be out of scope for this work.
> >
> >
> >
> > > It would be good to include a discussion between this work and work on direct image segmentation
> >
> > Common image segmentation architectures (like other task heads) use magnitudes more parameters and introduce their own inductive biases. This interferes with a comparison of backbones features. We clarified this by updating the introduction and the discussion of feature evaluation in the related work section.
> >
> >
> > > No discussion of simpler efforts on dense probing (e.g. dense probing in DINOv2)
> >
> > This is a valuable suggestion. We extended the discussion of other efforts to dense probing to include DINOv2 in the related work section.
> > Also note that the bicubic interpolation-based baseline in our experiments ("Comparison to dense readout baselines") is highly similar to the linear probing in the DINOv2 paper.
> >
> >
> > > There should be more discussion of ... 2) how the findings of this work relates to findings from prior work. **(C8)**
> >
> > We extended the discussion on the relation of our findings to prior work both in the results section and in the conclusion.

---

> > > ### Comment · Reviewer_AftN · 2025-06-23
> > > **Concerns are mostly addressed**
> > >
> > > As the revisions was not marked, it was a bit hard to see what has changed.  Nevertheless, I believe most of my concerns have been addressed.
> > >
> > > The main one that I think should be improved more is the following:
> > > - The connection between the text in Section 3, and Figure 1 (and the caption) can be further improved.  For instance, the figure would be much easier to follow if the caption started with describing what happens with the input image, instead of starting with the queries.   Additional symbols (e.g. x for the image, D' for the depth of the extracted features), indicating what part of the figure corresponds to M (I think that is the mask computation per head), T (before the MLP?), T' (after the MLP?), etc can really help tie the figure to the actual text.
> > >
> > > Minor updates
> > > - For bold headings that start a paragraph, there should be a period that separates the heading from the body of the paragraph (e.g. "Representation learning.", "Feature evaluation.", etc).  This should be fixed for all sections.
> > > - Section 4 only has one subsection (consider putting the text above 4.1 into a subsection)
> > > - Section 2: "Evaluations on features predate the deep learning era in computer vision." => There should be references to back up this statement, ideally with a bit of elaboration.
> > > - "3d" => "3D", "2d" => "2D"
> > > - A full proofreading pass is recommended as some sentences are awkward:
> > >   "Originating in natural language processing more recently these techniques were also applied in
> > > computer vision." => Should be rewritten - or just cut, as the information in this sentence isn't that important.

---

> > > > ### Author Response · Authors · 2025-06-26
> > > >
> > > > Thank you for your feedback.
> > > >
> > > > We decided not to highlight changes in the paper as OpenReview offers a tool for this (diff from original submission to latest update: https://openreview.net/revisions/compare?id=neMAx4uBlh&left=6PltqYS2oS&right=vlpyysYSbP&pdf=true&version=2)
> > > >
> > > > We improved the connection between Fig. 1 and the text by following your proposals and indcluded some additional changes (e.g. $\psi$ in Fig. 1 and Eq.1).
> > > > We also incorporated all your suggested minor changes.

---

### Review · Reviewer_tki8 · 2025-04-28

**Summary Of Contributions:**

This paper studies how to probe (efficiently assess) the feature quality given a backbone model in the context of dense prediction, which differs from conventional paradigms such as linear probing wherein the entire features are assessed for, e.g., image classification. The author proposes a method that is agnostic of the resolution of the feature.

The author has conducted rather comprehensive experiments to assess the validity of the proposed method on top of various backbones. The assessments comprise of instance awareness, local semantics and depth semantics.

**Audience:**

Yes

**Broader Impact Concerns:**

No broader impacts were identified.

**Claims And Evidence:**

Yes

**Requested Changes:**

The author could consider addressing the minor concerns I raised above.

**Strengths And Weaknesses:**

**Strengths**

* I think that the core idea is well-motivated and the designed module generally facilitates a fair comparison among backbones, given that in the conclusion the author proposes to maintain a leaderboard to further incorporate more, and new backbones I believe that this method could aid further research in this area.

* The execution of the idea is decent with comprehensive experiments.

**Weaknesses**

* I could not find major flaws in this work. Rather I am more curious as to whether the findings summarized in the conclusion (including identifying that the DINOv2 is of a higher performing model, while conventional backbones trained on classification are less effective, and vision-language models could better capture local semantics) are entirely new findings -- it seems that these topics may have been well covered in previous works, or are somewhat a general consensus among the research community, though as I do not directly work on the domain of, say, linear probing and related methods, I cannot definitively confirm this suspicion.

**Additional Comments**

As I do not directly work on this particular topic, I will later observe other reviewers' comments and potentially add more comments to my review.

---

> ### Author Response · Authors · 2025-06-12
>
> Thank you for your comments, we appreciate the positive assessment of our work.
> The finding of DINOv2 performing well is probably well known to most parts of the community. However, other findings, for example the representation of instances in self-supervised approaches or the strong correlation between backbone parameter size and segmentation and depth performance (this was added in the revision) are likely unknown.
> Beyond novel findings, the contribution of this paper is to establish a standardized method to evaluate the feature quality for dense prediction as direct as possible such that progress can also be tracked in the future.
>
> We look forward to further questions during the rebuttal period.

---

> > ### Comment · Reviewer_tki8 · 2025-07-24
> > **I have examined other reviewers' comments and author's response**
> >
> > After reading the comments of other reviewers and the authors' response I am leaning towards accepting because the authors have addressed the concerns candidly.

---

### Review · Reviewer_EEFu · 2025-05-29

**Summary Of Contributions:**

This paper proposes dense attentive probing to evaluate the feature quality of pre-trained backbones for dense prediction tasks like segmentation. The proposed probing is a parameter-efficient readout, which exploits a masked cross-attention layer with learnable mask sizes, to make dense predictions using arbitrary backbones independent of the size and resolution of their feature volume. Based on the proposed probing, this paper finds that DINOv2 outperforms all other backbones tested.

**Audience:**

Yes

**Broader Impact Concerns:**

There is no concerns on the ethical implications.

**Claims And Evidence:**

Yes

**Requested Changes:**

Please address the questions in Cons and revise the manuscript accordingly.

**Strengths And Weaknesses:**

Pros:
+ This paper proposes an efficient dense attentive probing strategy to evaluate the feature quality of pre-trained backbones for dense prediction tasks. Evaluating the feature quality of pre-trained backbone is an important and interesting topic to compare different self-supervised learning methods.
+ The dense attentive probing is adaptive to arbitrary backbones and independent of the size and resolution of their feature volume, showing a better trade-off between efficiency and efficacy compared with the other readout architectures.
+ This paper finds that DINOv2 outperforms all other backbones tested.

Cons:

1. The motivation in the Introduction is not easy to understand. For instance, please first define what is the "variable feature locality", then please mention why it is important for assessing feature quality. After that, please explain how you come up with the idea of using a learnable masking radius (rather than other strategies) to enforce the feature locality.

2. In the Introduction, please also clarify why the proposed readout can better handle the evaluation for SSL methods along three dimensions. It would be better to clearly present the connection between the readout and these three dimensions first.

3. In Fig. 1, how to construct the "Positions" of [H,W,2] as the input (i.e., how to obtain the values at each position in H*W*2)? What is the relation between the "Positions" and positional encoding (PE) of the grid coordinates p in Eq. (1)?

4. In Sec. 3, why can M_{ij} consider "spatial proximity"? The design of this mask is not easy to understand. Why can we directly add M to the cross-attention rather than multiplication?

5. Please give more details in the settings of Tab. 5, e.g., what does "only-mask" mean? Which one denotes only cross-attention without the mask M?

---

> ### Author Response · Authors · 2025-06-12
>
> Thanks for your valuable comments. Below we address your points.
>
>
> > The motivation in the Introduction is not easy to understand. For instance, please first define what is the "variable feature locality", then please mention why it is important for assessing feature quality. After that, please explain how you come up with the idea of using a learnable masking radius (rather than other strategies) to enforce the feature locality.
>
> As the backbones are trained with completely different methods, they may not encode local information solely at the corresponding spatial feature location, but rather distribute it across multiple locations. The learnable masking radius is not meant to enforce feature locality but to compensate it. We clarified these points in the introduction.
>
>
>
>
> > In the Introduction, please also clarify why the proposed readout can better handle the evaluation for SSL methods along three dimensions. It would be better to clearly present the connection between the readout and these three dimensions first.
>
> The dimensions represent different characteristics of dense features. The readout is designed specifically for dense prediction tasks and better suitable than competitive methods like linear evaluation because of (i) decoupling feature and output resolution, (ii) learning feature locality (at different levels per head) and (iii) using a small parameter budget. This argumentation was incorporated into the text.
>
>
>
>
> > In Fig. 1, how to construct the "Positions" of [H,W,2] as the input (i.e., how to obtain the values at each position in HW2)? What is the relation between the "Positions" and positional encoding (PE) of the grid coordinates p in Eq. (1)?
>
>
> The position grid consists of integer positions, i.e. $Q_{ij}=[PE(i), PE(j)]$, where $PE$ denotes the positional encoding function applied to the coordinates. We have extended both the text and Figure 1 to explain this relationship more clearly.
>
>
>
> > In Sec. 3, why can M_{ij} consider "spatial proximity"? The design of this mask is not easy to understand.
>
> We have revised the explanation of the masking mechanism in the paper.
> $M$ connects the query output space with the feature space. $i$ iterates over all (flattened) pixels of the output while $j$ iterates over the feature vectors. $M$ is built based on the distance between the spatial location of pixels $i$ and $j$.
>
> > Why can we directly add M to the cross-attention rather than multiplication?
>
> Adding the mask is common practice for modifying the attention scores (e.g. see [PyTorch's attention implementation](https://docs.pytorch.org/docs/stable/generated/torch.nn.functional.scaled_dot_product_attention.html) ). Due to the exponential functions in the softmax, the addition of m is turned into a multiplication: $softmax(a+m)= \frac{\exp(a)exp(m)}{\sum...}$.
>
>
>
> > Please give more details in the settings of Tab. 5, e.g., what does "only-mask" mean? Which one denotes only cross-attention without the mask M?
>
> We agree that the description of Tab. 5 lacks important information and revised the respective section in "Readout design".
> Mask-only means that query information is removed and thus attention is exclusively defined by the learned masks. The no-sigma variant is the one that uses cross-attention without masking. We renamed this to "no masking" for clarity.

---

> > ### Comment · Reviewer_EEFu · 2025-06-20
> > **Could you please upload a revision with changes highlighted?**
> >
> > Thanks for the response. However, it seems that the changes in revision are not highlighted. Could you please upload a revision with the changes highlighted to facilitate further discussion?

---

> > > ### Author Response · Authors · 2025-06-20
> > >
> > > We did not highlight changes because OpenReview has an internal compare tool for this purpose. You can find the changes here:
> > > https://openreview.net/revisions/compare?id=neMAx4uBlh&left=6PltqYS2oS&right=bTL5VaOIZW&pdf=true&version=2

---

> > > > ### Comment · Reviewer_EEFu · 2025-06-20
> > > > **Most of the concerns have been addressed**
> > > >
> > > > Thanks for the further clarification. The response has addressed most of my concerns. There are some minor comments for further improvement:
> > > >
> > > > For Q1, it would be better to use one or two sentences to clarify how the learnable masking radius compensates for feature locality in the Introduction. This can make the motivation clearer. In addition, it would be better to formally define the feature locality in the revision, e.g., we call xxx the issue of feature locality.
> > > >
> > > > For Q2, "the readout" (on page 2) is mentioned without an introduction to this concept. Please explain the concept “readout” before using the word "the".

---

> > > > > ### Author Response · Authors · 2025-06-26
> > > > >
> > > > > We appreciate your additional comments.
> > > > > The paper was revised. We elaborated on how the masking radius can compensate for different degrees of feature locality in the backbones. We decided against formally defining feature locality as it only serves as an intuitive motivation for our approach but we do not experimentally investigate feature locality in detail. This could be an interesting direction for follow-up work, though. We replaced the term readout for clarity.

---

> > > > > > ### Comment · Reviewer_EEFu · 2025-06-27
> > > > > > **The concern about motivation persists**
> > > > > >
> > > > > > Thanks for the response. However, the unclear definition of feature locality makes it more difficult to understand the motivation. Additionally, as pointed out by the authors, they do not experimentally investigate feature locality, which means that the intuitive motivation of feature locality lacks both theoretical and experimental support. This makes the motivation less convincing. Furthermore, it is still not clear how the learnable masking radius compensates for feature locality in the Introduction. Their relations should be clearly clarified to make the motivation easy to follow.

---

> > > > > > > ### Author Response · Authors · 2025-07-02
> > > > > > >
> > > > > > > We concur that elaborating on the definition of feature locality improves the paper. To this end, we revised the introduction section, including a more formal description of feature locality and added a new figure (Fig. 1) sketching the idea. This should also clarify the relation between learnable masking radius and feature locality in the backbones. Please let us know if there are some aspects that remained unclear.
> > > > > > >
> > > > > > > There is experimental support for feature locality. In the ablation, the variant without masking and the variant with a global mask ("no indiv. $\sigma$") for all attention heads perform worse than our model.

---

> > > > > > > > ### Comment · Reviewer_EEFu · 2025-07-03
> > > > > > > > **My concerns have been addressed**
> > > > > > > >
> > > > > > > > Many thanks for the further clarification. It addresses my concern. After revision, this paper has been improved and I believe that it is ready for acceptance.

---

### Author Response · Authors · 2025-06-12

We thank the reviewers for their valuable feedback, comments and questions. Based on the critique, we incorporated the following main changes in the paper:

**Presentation**
- Clarification of focus: probing/evaluation for dense prediction.
- Change of title to reflect the focus.
- Improved Fig. 1 (`EEFu`, `AftN`)
- Improved explanation of the function of masking (`EEFu`, `AftN`).
- Restructured results section for better flow. (`EEFu`, `AftN`)

**Related Work**
- Extended discussion of probing papers (`AftN`)
- Discussion of the linear evaluation of DINOv2 (`AftN`)

**Experiments**
- Additional ablations: number of heads, role of FFN and CNN. (`AftN`)
- Extended analysis of backbone comparison. (`AftN`)
- Rerun depth experiments with larger dataset.

Beside these points formulations were clarified at several places.

---

### Decision · Action_Editor_oJkj · 2025-08-14

**Recommendation:** Accept as is

**Additional Comments:**

All three reviewers recommended Accept. Minor writing and presentation improvements are still possible, while the scientific contribution is solid. There are additional efforts that authours can make,
1) Clarified definitions and motivations, improved figures, captions, and results structure.
2) Expanded related work and discussion of prior probing approaches.
3) Added ablations, further analysis on resolution effects, and clarified experimental setups.

**Audience:**

Yes

**Audience Explanation:**

Yes. Reviewers agreed the work would interest the TMLR community, especially those studying pretrained vision models, dense prediction tasks, and backbone evaluation protocols.

**Claims And Evidence:**

Yes

**Claims Explanation:**

Yes. All three reviewers agreed that the paper’s claims are backed by sound experiments and analyses. The authors conducted extensive evaluations on 18 vision backbones using the proposed dense attentive probing method, provided ablation studies, addressed reviewer concerns, and clarified motivations and methodology